# STAT3 Increases CVB3 Replication and Acute Pancreatitis and Myocarditis Pathology via Impeding Nuclear Translocation of STAT1 and Interferon-Stimulated Gene Expression

**DOI:** 10.3390/ijms25169007

**Published:** 2024-08-19

**Authors:** Tianming Liang, Zhipeng Zhang, Zhenxin Bai, Le Xu, Wei Xu

**Affiliations:** Jiangsu Provincial Key Laboratory of Infection and Immunity, Institutes of Biology and Medical Sciences, Soochow University, Suzhou 215123, China

**Keywords:** acute pancreatitis, CVB3, STAT3, STAT1, ISG

## Abstract

Acute pancreatitis (AP) is an inflammatory disease initiated by the death of exocrine acinar cells, but its pathogenesis remains unclear. Signal transducer and activator of transcription 3 (STAT3) is a multifunctional factor that regulates immunity and the inflammatory response. The protective role of STAT3 is reported in Coxsackievirus B3 (CVB3)-induced cardiac fibrosis, yet the exact role of STAT3 in modulating viral-induced STAT1 activation and type I interferon (IFN)-stimulated gene (ISG) transcription in the pancreas remains unclarified. In this study, we tested whether STAT3 regulated viral-induced STAT1 translocation. We found that CVB3, particularly capsid VP1 protein, markedly upregulated the phosphorylation and nuclear import of STAT3 (p-STAT3) while it significantly impeded the nuclear translocation of p-STAT1 in the pancreases and hearts of mice on day 3 postinfection (p.i.). Immunoblotting and an immunofluorescent assay demonstrated the increased expression and nuclear translocation of p-STAT3 but a blunted p-STAT1 nuclear translocation in CVB3-infected acinar 266-6 cells. STAT3 shRNA knockdown or STAT3 inhibitors reduced viral replication via the rescue of STAT1 nuclear translocation and increasing the ISRE activity and ISG transcription in vitro. The knockdown of STAT1 blocked the antiviral effect of the STAT3 inhibitor. STAT3 inhibits STAT1 activation by virally inducing a potent inhibitor of IFN signaling, the suppressor of cytokine signaling-3 ((SOCS)-3). Sustained pSTAT1 and the elevated expression of ISGs were induced in SOCS3 knockdown cells. The in vivo administration of HJC0152, a pharmaceutical STAT3 inhibitor, mitigated the viral-induced AP and myocarditis pathology via increasing the IFNβ as well as ISG expression on day 3 p.i. and reducing the viral load in multi-organs. These findings define STAT3 as a negative regulator of the type I IFN response via impeding the nuclear STAT1 translocation that otherwise triggers ISG induction in infected pancreases and hearts. Our findings identify STAT3 as an antagonizing factor of the IFN-STAT1 signaling pathway and provide a potential therapeutic target for viral-induced AP and myocarditis.

## 1. Introduction

Acute pancreatitis (AP) is a severe inflammatory condition of the pancreas, characterized by sudden-onset abdominal pain and elevated levels of pancreatic enzymes in the blood [1]. While gallstones and alcohol consumption are well-known etiological factors, emerging evidence suggests a potential link between viral infections and the development of AP [1,2]. Coxsackievirus B3 (CVB3), a member of the Picornaviridae family, has been implicated as a viral trigger for AP. CVB3 infection is commonly associated with viral myocarditis (VMC) and AP, among other clinical manifestations [3,4]. Despite the growing recognition of CVB3-induced AP, the underlying molecular mechanisms and how the virus modulates the antiviral type I interferon (IFN) response remain incompletely understood.

IFNs, composed of IFN-α and -β, constitute the primary defense mechanism against viral infections [5]. Antiviral functions of type I IFNs are mediated by binding with IFN receptors (IFNRs), recruiting JAK1-Tyk2 kinases for downstream signal transducer and activator of transcription (STAT) signaling and inducing IFN-stimulated genes (ISGs), which interfere with virus replication or trigger cell apoptosis to avoid viral spread [6]. Signaling through IFNRs triggers the activation (phosphorylation) of STAT proteins, including STAT1, STAT2 and STAT3. Phosphorylated STAT1 and STAT2 form heterodimers that translocate into the nucleus and bind to the IFN-stimulated response element (ISRE) promoter site, activating the transcription of hundreds of ISGs, including protein kinase R (PKR), mycovirus resistance genes (Mx), 2′-5′-oligoadenylate synthetase (OAS), inducible NO synthase (iNOS), interferon (IFN)-stimulated gene 56 (ISG56) (a positive regulator of CXCL10), ISG15 (conjugation and ISGylation of newly synthesized viral protein for interfering viral replication), viperin (broad-spectrum, virus-inhibitory protein) and CXCL10 (a STAT1-dependent gene), exhibiting diverse antiviral properties [7,8,9,10,11]. Activated STAT1 and STAT3 form a homo- or heterodimer and bind to the IFNγ-activated site, negatively regulating the type I IFN-mediated response.

Many viruses, including respiratory syncytial virus (RSV) [12] and severe acute respiratory syndrome coronavirus 2 (SARS-CoV-2), have evolved various countermeasures against the host IFN system, and particularly STAT1 expression and function [13]. During CVB3 infection, very low levels of IFNα/β and ISGs were induced in the cells and pancreases of mice [14,15,16]. The central mechanism lies in that CVB3 inhibits the RIG-I-like receptor (RLR) pathway upstream of TBK1 activation, and ISG induction. CVB3 2A(pro) directly degrades MDA5 and MAVS, and CVB3 3C(pro) targets RIG-I, MAVS, IRF7 and IRF9 for degradation, thereby nearly completely blocking the IFN I pathway from the beginning [17,18]. The search for drugs capable of targeting viral proteins implicated in both viral replication and IFN/STAT1 inhibition is important for the treatment of the most viral infections and for future viral pandemics.

During the initial 7 days of CVB3 infection, p-STAT1 was found in the hearts of mice and mainly stained positive in only a few infiltrating immune cells within the infiltrated clusters [19]; the IFNβ expression was below the detection levels in the pancreases and increased in the hearts of the mice at 8 days p.i. [16]. TLR3-PAR-1 cross-talk in mouse cardiac fibroblasts enhanced the induction of IFN-β, p-STAT1 and CXCL10 expression upon CVB3 infection [20]. Overall, CVB3 induces very low levels of IFN-β at early stages, indicating impaired STAT1 activation. In this study, our data also indicate that in the CVB3-infected pancreases, hearts and acinar cells, despite the increased STAT1 and moderately increased p-STAT1 expression, the relative p-STAT1/STAT1 expression failed to increase, and the nuclear translocation of p-STAT1 in the pancreas was heavily blunted, suggesting an impaired induction of the IFN I response.

STAT3, downstream of cytokine (IL-6, IL-10) or growth factor receptors, counteracts inflammation and promotes cell survival/proliferation and immune tolerance, while STAT1 inhibits proliferation and favors innate and adaptive immune responses [21]. STAT1 and STAT3 activations are reciprocally regulated and perturbations in their balanced expression or phosphorylation levels may re-direct cytokine/growth factor signals from inflammatory to anti-inflammatory [22]. In pathological contexts, the IL-10-mediated STAT activation pattern in high-IFNγ/IFNα-stimulated macrophages switches from being predominantly STAT3 to being mainly STAT1, leading to increased STAT1-STAT3 heterodimers, strengthening the proinflammatory functions of IL-10 [23,24].

A previous study found that CVB3 increased the p-STAT3 expression in the hearts of mice, exerting a cardiac apoptosis-protective role [25]. In CVB3-infected hearts and pancreases, we have found markedly increased p-STAT3 expression but not p-STAT1 expression. While STAT3 is activated by IFNs in various cell types, it has been reported recently to negatively regulate IFN-α-induced ISG expression and antiviral activity via the sequestering of STAT1, downregulating the expression of ISGF3 components, and blocking IFN-I signaling [26,27]. We speculate that the over-activation of p-STAT3 in the CVB3-infected pancreas and heart may function as a negative regulator of the type I IFN response via inducing dysfunctional STAT1 activation, and weak ISG production. Both the STAT1-antagonizing mechanism and the ISG-antagonizing mechanism need to be clarified.

Herein, we investigate the proviral activity of p-STAT3 and its antagonism against STAT1 signaling in CVB3-induced AP and VMC. Our results indicate that CVB3 increases early p-STAT3 over-activation in infected pancreases and hearts, which inhibits the IFN I response via blocking the nuclear translocation of p-STAT1 and ISG transcription. 

## 2. Results

### 2.1. CVB3 Upregulates p-STAT3 but Not p-STAT1 Expression in the Pancreases and Hearts of Mice

To assess the time course of the STAT3 and STAT1 activations in the pancreases and hearts after infection, C57BL/6 mice were i.p. injected with 10^3^ pfu CVB3. By day 7, the pancreases of the mice exhibited extensive immune infiltration as well as acinar cell necrosis histologically; while moderated immune infiltration was observed in the myocardia of mice (Figure 1A,B). The expression kinetics of p-STAT3 and p-STAT1 in the pancreases and hearts were determined. The pancreatic protein levels of p-STAT1 and STAT1 were unchanged during 7 days of infection. In contrast, the p-STAT3 expression increased from day 1, reached a maximum on day 3 p.i. (peaking time for CVB3 capsid VP1 expression), and declined by day 7 p.i., as demonstrated by the immunoblotting assay (Figure 1C). In the myocardia, the p-STAT3 but not p-STAT1 expression underwent a continuous increase from 1 to 7 days p.i. (Figure 1E). Next, in CVB3-infected murine acinar 266-6 and cardiomyocyte HL-1 cells, we also detected a marked increase in p-STAT3 in the 266-6 cells from 6 h, which reached a maximum at 12 h (Figure 1D) before declining afterwards, or in a continuous manner in the HL-1 cells from 6~24 h p.i. (Figure 1F). In contrast to that, the p-STAT1 levels in the cells remained unchanged or displayed a reduction after CVB3 infection. Taken together, CVB3 upregulated STAT3 but not STAT1 activation at an early phase in viral AP and VMC, indicating an impaired induction of the IFN I response.

### 2.2. CVB3 Infection Increases Nuclear Import of p-STAT3 but Blunts Nuclear Translocation of p-STAT1

Binding IFNs with IFNRs induces p-STAT1 and triggers the translocation of p-STAT1 heterodimers into the nucleus for binding ISREs and inducing ISG transcription [6,7]. To detect the nuclear translocation of STAT1 and STAT3, cytoplasmic and nuclear extracts of day 3 pancreases or hearts were prepared and subjected to immunoblotting analysis. We found a significant increase in the nuclear p-STAT3 expression and a marked decrease in the nuclear p-STAT1 expression. Accordingly, the cytoplasmic p-STAT3 level was decreased while p-STAT1 was increased at the same time point (Figure 2A,C). In consistency with this, immunoblotting of cytoplasmic and nuclear extracts of 0~24 h infected acinar 266-6 cells revealed an increased nuclear p-STAT3 expression accompanied by a decreased cytoplasmic p-STAT3 expression, while an increased cytoplasmic p-STAT1 expression was accompanied by a decreased nuclear p-STAT1 expression (Figure 2B). To further confirm that CVB3 failed to induce STAT1 activation, an immunofluorescence (IF) assay was performed. In mock-infected 266-6 cells, low levels of STAT1 and STAT3 were detected in both the cytoplasm and nucleus. In contrast to this, upon infection over time (6~24 h p.i.), STAT1 phosphorylation was enhanced but was mainly restricted in the cytoplasm, and, notably, almost no p-STAT1 was localized in the nucleus of infected 266-6 cells, while STAT3 phosphorylation and nuclear translocation were robustly increased by CVB3 during this period (Figure 2D). Collectively, CVB3 infection significantly increased the nuclear import of p-STAT3 but impaired the nuclear translocation of p-STAT1 in the pancreases and hearts.

### 2.3. STAT3 Inhibition Rescues Nuclear Translocation of p-STAT1 and Enhances ISG Expression in CVB3-Infected Cell

The phosphorylation of STAT1 and its nuclear translocation are the rate-limiting steps for the induction of ISGs [6]. Accumulating evidence has revealed STAT3 as a negative regulator of the type I IFN response [26,28,29]. Next, we tested to see whether STAT3 regulated STAT1 translocation to the nucleus. We pretreated cells with a STAT3 inhibitor, HIC0152 or Stattic, for 48 h, and then infected cells with CVB3 (MOI = 1) and measured the 12 h p-STAT1 amounts in the cytoplasmic and nucleic extracts. CVB3 induced reduced nucleic p-STAT1 in the 266-6 cells, which could be markedly rescued by HJC0152 or Stattic (Figure 3A). When the nuclear translocation of STAT1 was investigated by IF assay, we found that although viral infection increased the p-STAT1 expression, most of the p-STAT1 proteins were localized in the cytoplasm (almost undetectable nuclear STAT1) at 12 h p.i. When STAT3 was inhibited by HJC0152 or Stattic, a significant amount of p-STAT1 was remarkably translocated into the nucleus upon infection (Figure 3B). The p-STAT1 Fn/c values showed a significant increase in the HJC0152-treated group.

Next, we employed an ISRE luciferase reporter system to analyze the influence of the STAT3 inhibitor on the ISRE activity. It revealed that the CVB3-induced weak ISRE activity was significantly increased by the STAT3 inhibitor, as compared to that by DMSO (Figure 3C). To confirm the rescued p-STAT1 activation by the STAT3 inhibitor, the mRNA expressions of ISGs, including *lfit1/Isg56*, *Mxa*, *Rasd2*, *Isg15* and *Cxcl10*, were detected and showed a significant increase in HJC0152-/Stattic-treated cells (Figure 3D). WB analysis confirmed the upregulated expressions of MxA and viperin proteins after the HJC0152 treatment (Figure 3E). These data suggest that the nuclear translocation of p-STAT1 and ISG transcription were antagonized by the CVB3-increased p-STAT3, and the inhibition of STAT3 restored the nuclear translocation of p-STAT1 and ISG transcription.

### 2.4. STAT3 Inhibitor Restricts CVB3 Replication via STAT1-Dependent Way

To determine whether STAT3 inhibition promotes viral clearance, lentivirus-vectored shSTAT3 was transfected into 266-6 cells 48 h before CVB3 infection, achieving a 90% STAT3-inhibiting efficiency. Compared to scrambled shRNA, shSTAT3 significantly reduced the VP1 expression (Figure 4A). When 266-6 cells were treated with a STAT3 inhibitor, HJC0152 or Stattic, before CVB3 infection, both STAT3 inhibitors dose-dependently decreased the p-STAT3 and VP1 expression (Figure 4B). An immunofluorescence (IF) assay revealed that 12 h after eGFP-CVB3 (MOI = 1) infection, HJC0152 markedly reduced the eGFP+CVB3 progeny production compared to the DMSO treatment (Figure 4C). To clarify that STAT3-enhanced CVB3 replication is mediated through modulating STAT1 signaling, the STAT1 expression was knocked down by lentivirus–shRNA transfection 48 h before infection. Compared to scrambled shRNA, shSTAT1 restored the VP1 expression that was decreased by HJC0152 (Figure 4D). The inhibitory effect of HJC0152 on the supernatant viral titer was also abolished by the STAT1 knockdown (Figure 4E). These data indicate that STAT3 inhibition effectively restricted viral replication via increasing STAT1 signaling.

### 2.5. CVB3 VP1 Induces STAT3 Activation, Which Induces Expression of Negative Regulator, SOCS3, to Block pSTAT1/ISG Expression

We confirmed that CVB3 dose-dependently promoted p-STAT3 expression in the 266-6 cells (Figure 5A). To identify the viral capsid protein involved in triggering the activation of STAT3, we transfected HEK293T cells with plasmids encoding viral capsid protein VP1-VP4 48 h before CVB3 infection and screened for their ability to induce p-STAT3 using a Western blot assay. Compared to the vector-treated cells, of all these proteins, over-expressed VP1 markedly upregulated the p-STAT3 expression (Figure 5B). To confirm this effect, 0.5~1 μg of pVP1-Flag was transfected into 266-6 cells 48 h before CVB3 infection, and it was found that the VP1 protein dose-dependently increased the p-STAT3 activation (Figure 5C). Taken together, CVB3 capsid protein VP1 directly activates STAT3 activation in the pancreas at an early phase of infection.

Distinct genes belonging to the Suppressor of Cytokine Signaling (SOCS) family are induced as immediate early genes downstream of STATs and inhibit STAT phosphorylation by different mechanisms in a negative-feedback loop [30]. p-STAT3 is associated with the induction of the expression of SOCS3, a professional phosphorylation inhibitor of STATs [31]. Here, we wondered whether SOCS was involved in the inhibitory mechanism of p-STAT3 on p-STAT1. Acinar 266-6 cells were transfected with scrambled shRNA or shRNA-STAT3 48 h before CVB3 infection, and the relative mRNA expression of the STAT1 negative regulators SOCS1, SOCS3 and protein inhibitor of activated STAT (PIAS) 1/3 (nuclear STAT binder) was evaluated. Compared to the scrambled shRNA treatment, shRNA-STAT3 significantly increased the mRNA expression of SOCS3 (Figure 5D), concomitantly with a robust induction of the SOCS3 protein level (Figure 5E). To further assess the effect of SOCS3 in regulating the virus-induced suppression of p-STAT1, wild-type (WT) and shRNA-SOCS3-transfected 266-6 cells were infected with CVB3. Cell lysates were assessed for p-STAT1. In contrast to no STAT1 activation in CVB3-infected WT cells, the infection of SOCS3 knockdown cells resulted in strongly elevated p-STAT1 (Figure 5F). To answer the question as to whether enhanced p-STAT1 in SOCS-3 knockdown cells would also lead to the enhanced expression of ISGs, total RNA was isolated at different time points p.i. from infected cells and monitored for the induction of ISGs, Mxa, Isg15 and Cxcl10 (Figure 5G). The mRNA levels of all three representative ISGs were elevated in the SOCS3 knockdown versus WT cells at almost every time point during the course of infection. Meanwhile, the knockdown of SOCS3 resulted in decreased viral replication (Figure 5F). Taken together, the data indicate that, in the absence of SOCS3, infection leads to a stronger activation of STAT1, resulting in the enhanced expression of ISGs and reduced virus propagation.

### 2.6. Therapeutic STAT3 Inhibitor Treatment Mitigates AP and VMC Pathology via Rescuing ISG Induction

To evaluate the treating efficacy of the STAT3 inhibitor against CVB3-induced AP and VMC, mice were i.p. treated with 12.5 mg/Kg HJC0152, 1 d before and 1 d after 10^3^ pfu CVB3 infection on day 0 (Figure 6A). By 7 dpi, compared to non-treated mice, the HJC0152-treated mice had a higher survival rate and lower weight loss, indicating improved disease progression (Figure 6B). We next investigated whether the improved disease progression of the STAT3 inhibitor-treated mice was accomplished by a reduced viral titer. Western blots on day 3 tissue homogenates revealed a marked reduced VP1 expression in the pancreases and hearts of the HJC0152-treated mice compared to the non-treated mice (Figure 6C). This result was consistent with an enhanced expression of pancreatic IFN-α and IFN-β by day 3 p.i. (Figure 6D), along with the significantly increased expressions of pancreatic ISGs (*Oas1*, *Mxa*, *Rasd2*, *Isg15* and *Cxcl10*) (Figure 6E). Immunoblotting of day 3 pancreas lysates revealed the elevated expressions of MxA and viperin protein after the HJC0152 treatment (Figure 6F). Compared to the mock treatment, the STAT3 inhibitor substantially reduced the necrosis and immune infiltration in the infected pancreases and inflammatory injury in the hearts of the mice on day 7 p.i. (Figure 6G). The reduction in AP pathology by HJC0152 was also reflected in the markedly reduced levels of proinflammatory cytokines (TNFα, IL-6, IL-1β, IL-17A) compared to the non-treated pancreas (Figure 6H). Collectively, our data indicate that STAT3 inhibition mitigates viral-induced AP and VMC pathology via restoring the type I IFN response and reducing tissue inflammation.

## 3. Discussion

The unbalanced expression and/or activation of STAT1 and STAT3 has been linked to several pathological conditions. An impaired type I IFN response plays a key role in CVB3 pathogenesis. Our findings reveal a novel IFN-antagonism mechanism by CVB3 to trigger p-STAT3 over-activation, which impedes the nuclear translocation of p-STAT1 in the pancreas and heart at an early phase via inducing SOCS3, thus impairing a timely induction of the IFN response and ISG transcription that is vital for viral clearance. 

Type I interferons (IFNs) serve as the first line of defense against viral infections. They induce the production of a cascade of antiviral proteins in infected cells, thereby impeding viral replication and dissemination. Additionally, they activate natural killer cells and other immune cells, enhancing their ability to recognize and eliminate virus-infected cells [6,7]. In the ongoing battle of host–virus co-evolution, viruses have evolved intricate mechanisms to counteract host defense systems, often by subverting IFN secretion and limiting the production of antiviral mediators [32,33,34,35,36]. For instance, the nucleocapsid protein of SARS-CoV-2 inhibits the aggregation of MAVS, a crucial adaptor in IFN signal transduction, thereby impairing IFN-mediated antiviral immunity [33]. The nucleoprotein of the influenza A virus hampers the RIG-I-mediated innate sensing of RNA viruses [32], while the Ebola virus encodes the VP35 protein to antagonize RIG-I signaling, facilitating its persistence [35]. Our results, along with those of others, demonstrate that CVB3 vividly suppresses the IFN response by affecting MAVS and MDA5 [15,17,18,37,38]. Hence, a logical approach for antiviral therapeutics is to antagonize the viral mechanism to enhance the expressions of ISGs and IFNs to combat viral infection. Considering the crucial role of STAT1 in the IFN response, viruses have evolved different mechanisms to block STAT1 expression and function [39]. SARS-CoV-2 3CLpro restricts IFN induction by reducing the K63-linked ubiquitination of RIG-I and the Mpro-mediated degradation of STAT1 [40]. Hepatitis C virus (HCV) NS5A inhibits IFNα signaling through the suppression of STAT1 phosphorylation in hepatocytes [41]. Enteroviruses have also evolved a series of strategies to blunt STAT1 activation. EV71 Viral 2B induces the degradation of karyopherin-α1 (KPNA1), a component of the p-STAT1/2 complex, and thus suppresses type I interferon (IFN) responses [42].

STAT3 is a member of the STAT protein family involved in antiviral defense. STAT3 protein, phosphorylated and activated by IL-6 and IL-10, holds a pivotal function in cell signaling and transcriptional regulation [43,44]. STAT3 plays a crucial and complex role in the interaction between viruses and the host immune system [45], particularly receiving significant attention after the COVID-19 pandemic [46]. STAT3 has a protective role at the early stage of several viral infections. STAT3 is required for an optimal IFN response, as well as the splenic expression of IFN-α and ISGs to Herpes Simplex Virus-1 (HSV-1). Myeloid-specific STAT3 KO mice were more susceptible to HSV-1, as marked by their higher viral loads and impaired NK and CD8^+^T cell activation [47]. Enterovirus 71 (EV71)-induced p-STAT3 exerts anti-EV71 activity [48]. STAT3 exhibits antiviral activity in influenza virus (IAV) infection by activating ISG (PKR, OAS2, ISG15 and MxB) expression [49]. However, more and more evidence shows that STAT3 activation is associated with increased viral replication. The epidermal growth factor receptor (EGFR) inhibitor erlotinib inhibits Hepatitis B virus (HBV) replication via the downregulation of p-STAT3 in HBV-transfected HepG2.2.15 cells [50]. Influenza A virus infection activates STAT3 to enhance SREBP2 expression, cholesterol biosynthesis and virus replication [51]. Varicella-Zoster Virus (VZV) inhibits STAT1 activation and triggers p-STAT3 in vitro and in human skin xenografts in SCID mice in vivo; p-STAT3 inhibitors restrict VZV replication and skin malignant transformation [52]. Human Cytomegalovirus (HCMV) IE1 protein interacts with and promotes STAT3 nuclear accumulation, which increases viral replication [53]. African swine fever virus (ASFV) CD2v activates the JAK2/STAT3 pathway at the early stage and inhibits the apoptosis of infected cells, thereby promoting viral replication [54]. Furthermore, JAK2/STAT3 signaling contributes greatly to the cytokine storm seen in severe COVID-19 via inducing the Th1/Th17 immune response, hyperactivating STAT3 while impairing STAT1 function, thereby suppressing the antivirus interferon response and coagulopathy [55]. Although a previous report suggested a cardiac fibrosis-protective role of STAT3 in CVB3-induced VMC [56], the exact role and mechanism signaling of STAT3 in CVB3 infection and tissue pathology remain unclear [20]. In the current study, we found that CVB3 capsid protein VP1 induced p-STAT3 hyperactivation in the pancreases and hearts of mice at an early phase of infection. The notably increased activation and nuclear translocation of STAT3 promoted viral replication and tissue histopathology. The significant finding of this study is that we define STAT3 as a negative regulator of the IFN I response and ISG induction via impeding the nuclear translocation of p-STAT1 via inducing SOCS3.

STAT3 exerts multifaceted control over the expression and signaling of IFN-I [28]. There are three potential mechanisms by which STAT3 may block the expression and nuclear translocation of p-STAT1 and antagonize IFN-I signaling: (1) To compete for STAT1 association and DNA binding [27]. STAT3 competes with STAT1 for KPNA1 binding for nuclear transportation to inhibit the IFN I pathway in EV71 infection [29]. STAT3 binds to phospholipid scramblase 1/2 (PLSCR2) to suppress ISG expression and antiviral activity via interfering with STAT1 activation and nuclear translocation and ISGF3 complex assembly [57]. (2) STAT3 induces the expression of the negative regulator SOCS3 to block IFNα-induced p-STAT1/ISG expression and IFN-I signaling. SOCS3 is well known as a feedback inhibitor of the JAK/STAT3 signaling pathway. Heightened STAT3 activity prompts the induction of suppressive SOCS family members, which dampen IFN-I signaling [58], causing an impaired antiviral response in IAV infection [59]. (3) STAT3 heterodimerization with STAT1 attenuates STAT1 transcriptional activity on inflammatory gene promoters. Thus, the (patho)physiological regulation of the relative expressions of p-STAT1 and p-STAT3 will determine the nature of type I IFNs during viral-induced inflammatory diseases. 

The dysregulation of type I IFN signaling due to abnormal STAT3 activation not only promotes viral persistence but also exacerbates tissue damage and inflammation. STAT3 signaling in glia cells participates in the progression of EV71-induced neurogenic pathogenesis [29]. In the current study, viral-induced STAT3 hyperactivation in the pancreases and hearts contributed to increased viral replication and tissue inflammatory injury (Figure 6B,G,H). The IL-6/JAK2/STAT3 signaling pathway is vividly involved in the pancreatic intraepithelial neoplasia (PanIN) and the development of pancreatic cancer [60]. STAT3 orchestrates a complex interplay with IFN-I regulation through various mechanisms contingent upon the cellular context, immune milieu and pathological conditions. Targeting the STAT3 pathway to restore type I IFN responsiveness possesses therapeutic potential for enhancing host antiviral defenses and mitigating viral-induced tissue pathology. 

Our findings identify STAT3 as a negative regulator of STAT1-mediated IFN signaling during CVB3 infection of the pancreas and heart. Hyperactivation of p-STAT3 impedes the nuclear translocation of p-STAT1 and ISG transcription in pancreatic acinar cells and cardiomyocytes via inducing SOCS3. The pharmaceutical inhibition of STAT3 exhibits a therapeutic effect against viral pancreatitis and myocarditis via efficient, early viral clearance. We suggest the therapeutic potential for targeting early STAT3 in the future treatment of CVB3-induced AP and VMC.

## 4. Materials and Methods

### 4.1. Animals, Virus and Cells

Male C57BL/6J mice were purchased from Shanghai Laboratory Animal Center (Shanghai, China) and housed in a pathogen-free facility. CVB3 (Nancy strain) was a gift from Prof. Y. Yang (Key Laboratory of Viral Heart Diseases, Zhongshan Hospital, Shanghai, China) and was propagated in a Hela cell monolayer. Acinar cell line 266-6, a gift from Prof. H. Liu (IBMS, Soochow University, Suzhou, China), and cardiomyocyte cell line HL-1, a gift from Prof. T. Peng (University of Western Ontario, London, ON, Canada), were cultured in Dulbecco’s modified Eagle medium (DMEM) (HyClone) with 10% fetal bovine serum (FBS) and penicillin and streptomycin (100 U/mL) at 37 °C in an incubator containing 5% CO2. p-VP1/VP2/VP3/VP4-Flag plasmids were a gift from Prof. Sidong Xiong (IBMS, Soochow University, Suzhou, China).

### 4.2. CVB3-Induced Acute Pancreatitis and Histopathology Evaluation

Male C57BL/6 mice of 6 wk of age were intraperitoneally (i.p.) inoculated with 10^3^ pfu CVB3 in 100 μL PBS. The weights of the mice were recorded daily from day 1 to day 7. The pancreases, hearts and livers were removed at 0-, 1-, 3- or 7-days postinfection (p.i.), half for detecting the viral load and inflammatory cytokine levels and half for the histopathology. Briefly, tissues were longitudinally cut, fixed in 10% phosphate-buffered formalin, embedded in paraffin, sectioned at a 5 μm thickness, and stained with H&E. Images were captured randomly by a Nikon Eclipse TE2000-S microscope (Nikon, Tokyo, Japan); the inflammatory and edema/necrosis scores were quantified by a semiquantitative scale: 1 represented 25% of the tissues affected; 2 represented 25–50% involvement of the tissue; and 3 represented 50% involvement of the tissue.

### 4.3. Cell Treatment and Virus Infection

Cells were seeded at a density of 10^4^ cells/well. After overnight culture, 266-6 cells were infected with CVB3 (MOI = 1) for 1 h and, after washing, cells were collected after 6, 12, 18 and 24 h. In experiments where STAT3 inhibitors were used, cells were pretreated with 10 μM HJC0152 (MedChemExpress, HY-100602, Monmouth Junction, NJ, USA) or 2.5 μM Stattic (MedChemExpress, HY-13818) dissolved in DMSO (Sigma, St. Louis, MO, USA) for 48 h and then exposed to CVB3 infection. For transfection, 1 μg p-VP1-Flag, p-VP2-Flag, p-VP3-Flag and p-VP4-Flag plasmids or 0, 0.5, 0.75 and 1 μg p-VP1-Flag plasmid were transfected into 266-6 cells 48 h before CVB3 infection.

### 4.4. Plaque Assay

The intracellular viral load was titrated through a plaque-forming assay. Cells were washed with PBS, followed by subjecting them to three freeze–thaw cycles and sonication to release viral particles. Cellular debris was removed by centrifugation at 300× *g* for 10 min. Briefly, the viral supernatant was applied to 95% confluent Hela cells and incubated for 1 h. After washing with PBS, the infected cells were covered with 0.6% agarose–DMEM and further cultured for 72 h. Following the removal of agarose, the monolayer was fixed with 4% paraformaldehyde for 2 h. Finally, staining with 0.1% crystal violet was carried out for 1 h to visualize and count the plaques.

### 4.5. Western Blot

Pancreas and heart tissues (5 mg) were homogenized in 100 μL RIPA lysis buffer (TargetMol, C0045, Boston, MA, USA) containing PMSF, digested for 20 min at 4 °C to obtain lysates and then centrifugated at 2000× *g* for 15 min at 4 °C. The protein content was quantified by a BCA Protein Assay Kit (Pierce, 23227, Rockford, IL, USA). Protein extracts were separated on a 12% polyacrylamide gel and transferred to PVDF membranes using the Trans-Blot^®^ Turbo™ Transfer Buffer and System. The membranes were blocked with 5% non-fat dry milk in Tris-buffered saline containing 0.2% Tween 20 (TBST) for 1 h at room temperature, and were then incubated with anti-VP1 (Mediagnost; M47, Reutlingen, Germany), anti-p-STAT3 (Tyr705, CST, D3A7, Waltham, MA, USA), anti-STAT3 (CST, 124H6), anti-STAT1 (CST, D1K9Y), anti-p-STAT1 (CST, Tyr701, 58D6), anti-MX1 (Abcam, ab197205, Cambridge, UK), anti-Viperin (Abcam, ab107359), anti-SOCS3 (Abcam, ab280884), anti-GAPDH (CST, D16H11) or anti-Histone H3 (CST, 1B1B2) in 1% non-fat milk–TBST overnight at 4 °C. Protein bands were probed with HRP-conjugated secondary Abs, and the target proteins were visualized using the SuperSignal West Femto kit (Pierce, 34094). Densitometric measurements were performed using ImageJ software (version 2.1) and were normalized to loading controls.

### 4.6. Enzyme-Linked Immunosorbent Assay (ELISA)

Levels of mouse TNF-α, IL-1β, IL-6 and IL-17A in pancreas homogenates were analyzed by an ELISA kit (Invitrogen, Waltham, MA, USA) according to the manufacturer’s instructions. Color density was measured at 450 nm and 570 nm using BioTek Synergy H4 (Winooski, VT, USA).

### 4.7. Nucleic and Cytoplasmic Fraction Purification

Nuclear and cytoplasmic proteins were purified using the Nuclear and Cytoplasmic Protein Extraction Kit (Beyotime, Haimen, China).

### 4.8. Luciferase Reporter Assay

For ISRE reporter activity detection, pISRE-Luc and a transfection control, pRL-TK, were co-transfected with siRNA-STAT3 into cells; 24 h post-transfection, cells were treated with HJC0152 (10 μM) for 48 h, Stattic (2.5 μM) for 24 h or IFN-α (1000 IU/mL) as a positive control for 8 h; following 24 h of CVB3 infection, cell lysates were subjected to dual luciferase activity analysis by the Dual Luciferase Assay System (Promega, Madison, WI, USA). Data were normalized to Renilla luciferase activity.

### 4.9. shRNA Knockdown

Knockdown of STAT1, STAT3 and SOCS3 was achieved using shSTAT1, shSTAT3 and shSOCS3 lentiviral particles and following the manufacturer’s protocol. Acinar 266-6 or HEK293T cells were transfected with 2 μg packaging plasmid, 200 ng of VSV-G envelope plasmid and 2 μg of pLKO shRNA (shRNA-STAT1, -STAT3, -SOCS3) plasmid using Lipofectamine 2000 (Invitrogen, Carlsbad, CA, USA). The viral supernatant was filtered and used to infect cells overnight with 8 μg/mL polybrene. 

### 4.10. Quantitative Real-Time PCR (qPCR)

Pancreatic tissues (10 mg) or cells were treated with 1 mL of RNAiso Reagent (AG RNAex Pro RNA, AGbio, Westminster, CO, USA). Total RNA was extracted using the trichloromethane–isopropanol method and dissolved in RNase-free water. An amount of 1 μg total RNA was reverse transcribed into cDNA (Takara, Shiga, Japan), and contaminating DNA was removed with genomic DNA Eraser (Takara). qPCR was performed with a Bio-Rad iCycler using the Taq Perfect real-time kit (Takara) and the following primers (Tsingke Biotech, Beijing, China): *mxa* For: 5′-GCAGAAGGTCAGAGAGAAGG-3′, Rev: 5′-AGGGATGTGGCTGGAGATG-3; *oas1* For: 5′-GGGATTTCGGACGGTATTG-3′, Rev: 5′-CTCAGCCTCTTGTGCCAGC-3; *ifit1* For: 5′-AGAAGCAGGCAATCACAGAAAA-3′, Rev: 5′-CTGAAACCGACCATAGTGGAAAT-3; *rsad2* For: 5′-CGTGGAAGAGGACATGACGGAAC-3′, Rev: 5′-CCGCTCTACCAATCCAGCTTC-3; *Isg15* For: 5′-GGTGTCCGTGACTAACTCCAT-3’, Rev: 5′-TGGAAAGGGTAAGACCGTCCT-3′; *Cxcl10* For: 5′-CCAAGTGCTGCCGTCATTTTC-3’, Rev: 5′-GGCTCGCAGGGATGATTTCAA-3; *Ifn-α* For: 5′-CTTCCACAGGAT ACTGTGTACCT-3′, Rev: 5′-TTCTGCTCTGACCACCCTCCC-3′; *Ifn-β* For: 5′-CCTGTGTGATGCAGGAACC-3′, Rev: 5’-TCACCTCCCAGGCACAGA-3′; *Socs1* For: 5′-CTGCGGCTTCTATTGGGGAC-3′, Rev: 5′-AAAAGGCAGTCGAAGGTCTCG3′; *Socs3* For: 5’-ATGGTCACCCACAGCAAGTTT-3′, Rev: 5′-TCCAGTAGAATCCGCTCTCCT-3′; *Pias1* For: 5′-GCGGACAGTGCGGAACTAAA-3′, Rev: 5′-ATGCAGGGCTTTTGTAAGAAGT-3′; *Pias3* For: 5′-GAAGGAGGCATCAGAGGTTTG-3′, Rev: 5′-AGACAGGAAATCACTGCCCA-3′. Relative amount of specific mRNA was normalized to that of GAPDH. The relative quantification was calculated using the 2^−ΔΔCt^ cycle threshold method.

### 4.11. STAT3 Inhibitor Treatment Experiment

Male C57BL/6 mice were i.p. injected with 12.5 mg/Kg of HJC0152 1 day before and 1 day after infection with 10^3^ pfu CVB3 on day 0, and mice in the sham group were i.p. injected with an equivalent volume of vehicle. The pancreas was removed at 3 or 7 d p.i. to evaluate vital titers and histopathology of tissues.

### 4.12. Immunofluorescent Staining

Cells were treated with 10 μM HJC0152 and then infected with eGFP-CVB3 at MOI = 1. Fluorescence microscopy was performed on cells at 12 h p.i. Cells were fixed with 4% paraformaldehyde, permeabilized with 0.5% permeabilization agent and the nucleus was stained with DAPI, followed by thorough washing. Images captured under visible light showed similar cell densities at 12 h p.i. They were obtained using an A1 confocal microscope (Nikon).

### 4.13. Confocal Laser Scanning Microscopy

To monitor the translocation of STAT1 and STAT3, cells were seeded onto glass coverslips and cultured overnight, and then cells were pretreated with HJC0152 (10 μM) for 48 h. After 1 h infection with CVB3 (MOI = 1), cells were washed and treated with HJC0152 for 12 h. Cells were fixed with 4% paraformaldehyde and permeabilized with 0.5% Triton X-100 for 15 min. After being washed and blocked with BSA, the cells were incubated with primary anti-p-STAT1 (CST, 58D6) or anti-p-STAT3 (CST, D3A7) Abs for 2 h, followed by secondary antibodies for 50 min. The nucleus was stained with DAPI followed by thorough washing. Slides were monitored by a Carl Zeiss LSM 510 meta confocal microscope (Oberkochen, Germany).

### 4.14. Statistical Analysis

Multiple-group comparisons were performed by one way ANOVA, followed by Bonferroni post-tests. Continuous homoscedastic parametric comparisons of two groups were conducted by the two-tailed Students’ *t*-test. Histopathology was assessed by Mann–Whitney U test, and survival curves were compared by the log-rank test. All statistical analyses were conducted using GraphPad Prism 8.0 (GraphPad Software, San Diego, CA, USA), with the significance threshold set at 0.1 or better.

## Figures and Tables

**Figure 1 ijms-25-09007-f001:**
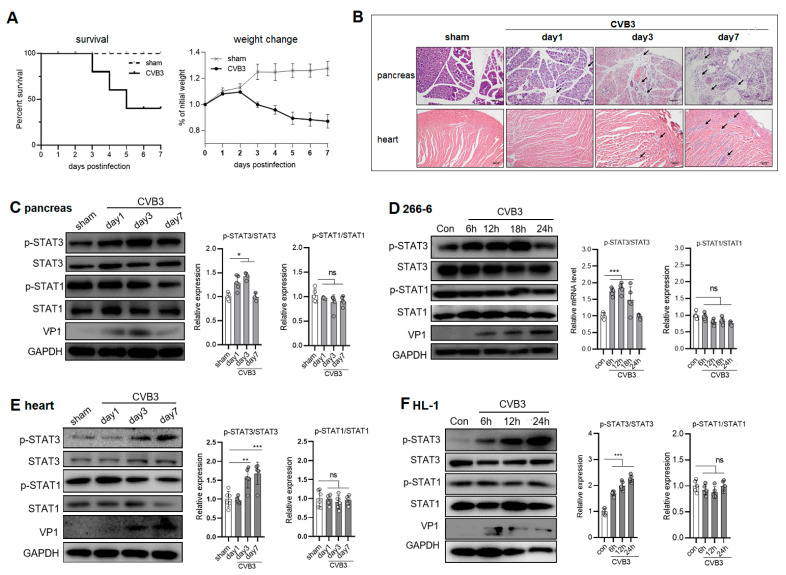
CVB3 increased p-STAT3 but not p-STAT1 expression in infected pancreases and hearts of mice. C57BL/6 mice were i.p. injected with 10^3^ pfu CVB3. (**A**) Body weight change and survival curve of mice (*n* = 7) infected with CVB3 were monitored by day 7 p.i. (**B**) Representative hematoxylin–eosin (H&E)-stained pancreas and heart sections from infected mice. Arrows indicate lymphocyte infiltration. Scale bar: 100 µm. (**C**–**F**) Protein levels of p-STAT1, STAT1, p-STAT3, STAT3 and VP1 in pancreas (**C**) or heart (**E**) lysates on 0, 1, 3 and 7 days p.i. or in murine acinar 266-6 (**D**) or cardiomyocyte HL-1 (**F**) cells infected with CVB3 (MOI = 1) at 6, 12 and 24 h p.i. were determined by immunoblotting analysis. GAPDH was probed as a protein loading control. STAT3/STAT1 relative expressions were quantitated by densitometric analysis and normalized to GAPDH, and they are presented as fold changes compared with the control. *, *p* < 0.1; **, *p* < 0.01; ***, *p* < 0.001. ns, no significance.

**Figure 2 ijms-25-09007-f002:**
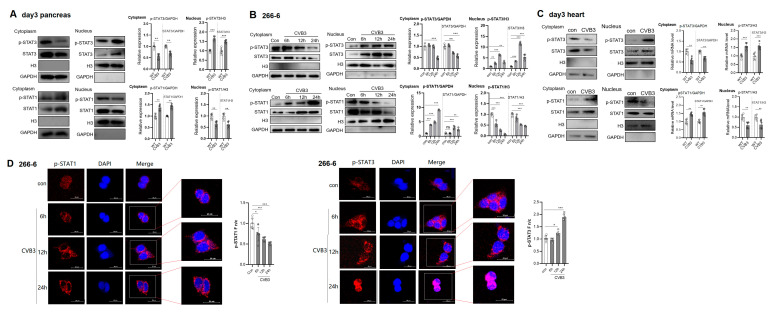
CVB3 infection increases nuclear import of p-STAT3 but blunts nuclear translocation of p-STAT1. (**A**,**C**) Nuclear and cytoplasmic extracts from day 3 mouse (**A**) pancreases and (**C**) hearts were prepared and subjected to immunoblotting with anti-p-STAT3, anti- STAT3, anti-p-STAT1 and anti-STAT1 antibodies. (**B**) Nuclear and cytoplasmic extracts were prepared from CVB3-infected 266-6 cells and measured by immunoblotting. Protein bands were determined via densitometric analysis, normalized to GAPDH and are presented as fold changes compared with the control. (**D**) CVB3 infection increased nuclear import of p-STAT3 but blunted nuclear translocation of p-STAT1 in vitro. Acinar 266-6 cells were mock-infected or infected with CVB3 and were fixed at 0, 6, 12 and 24 h p.i. p-STAT1 and p-STAT3 were analyzed with specified antibodies by confocal microscopy. p-STAT1 or p-STAT3 nuclear fluorescence (Fn), cytoplasmic fluorescence (Fc) and background fluorescence (Fb) were analyzed by ImageJ software and calculated using the equation Fn/c = (Fn − Fb)/(Fc − Fb). Results are from a single assay representative of three independent analyses. Data are shown as the mean ± SEM. *, *p* < 0.1; **, *p* < 0.01; ***, *p* < 0.001.

**Figure 3 ijms-25-09007-f003:**
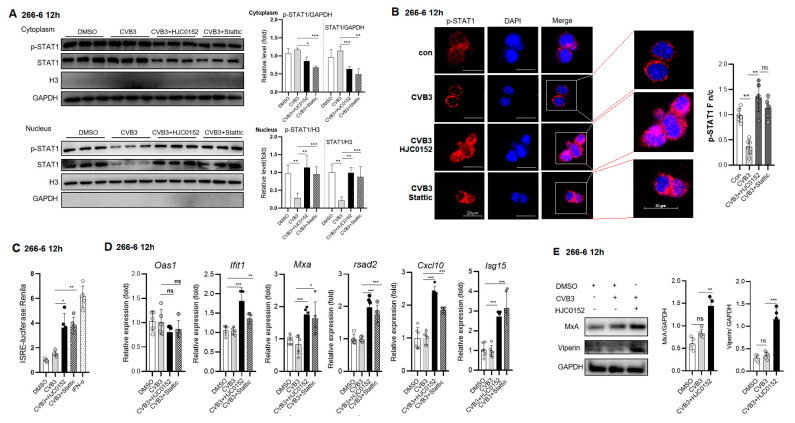
STAT3 inhibition rescues nuclear translocation of p-STAT1 and enhances ISG expression. Acinar 266-6 cells were treated with HJC0152 (10 μM) for 48 h or Stattic (2.5 μM) for 24 h and then infected with CVB3 (MOI = 1) for 12 h. (**A**) Nucleic or cytoplasmic p-STAT1 and STAT1 expressions were analyzed by immunoblotting analysis. Protein levels were quantitated by densitometric analysis and normalized to GAPDH, and they are presented as fold changes compared with the control. (**B**) IF assay to analyze nuclear translocation of p-STAT1 in STAT3 inhibitor-treated cells. Acinar 266-6 cells were mock-infected or infected with CVB3 and fixed 12 h p.i. The p-STAT1 expression and localizations were analyzed by confocal microscopy. The ratio of STAT1 nuclear fluorescence (Fn) to cytoplasmic fluorescence (Fc) was calculated and is shown. Data are shown as the mean ± SEM from three independent analyses. (**C**) STAT3 inhibitor increased ISRE activity in CVB3-infected cells. Acinar 266-6 cells were co-transfected with ISRE-Luc and Renilla for 24 h and then were treated with HJC0152 (10 μM) for 48 h before CVB3 infection. We subjected 12 h cell lysates to dual luciferase activity determination. (**D**) mRNA levels of ISGs (OAS, MXA, ISG15, ISG56, Viperin and CXCL10 and GAPDH) from HJC0152-treated cells were analyzed by RT-QPCR. (**E**) Immunoblotting analysis of MxA and Viperin proteins in HJC0152-treated cells at 12 h p.i. ns, no significance. *, *p* < 0.1; **, *p* < 0.01; ***, *p* < 0.001.

**Figure 4 ijms-25-09007-f004:**
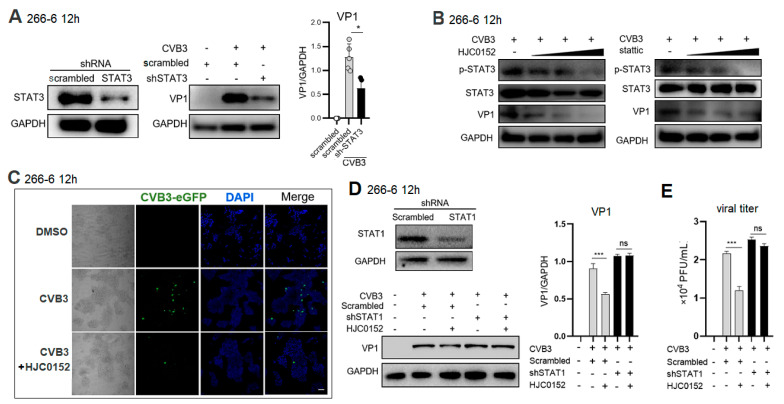
STAT3 inhibition restricts CVB3 replication via increasing STAT1 activation in pancreatic cells. (**A**) Acinar 266-6 cells were transfected with scrambled shRNA or STAT3-specific shRNA for 48 h. STAT3 protein level was detected by immunoblotting assay. (**B**) VP1, p-STAT3 and STAT3 expressions at 12 h p.i. were determined by Western blot assay in 266-6 cells treated with HJC0152 (2.5~10 μM) 48 h before or with Stattic (1~5 μM) 24 h before CVB3 infection (MOI = 1). (**C**) Fluorescence microscopy of 266-6 cells infected with eGFP-CVB3 and treated with 10 μM HJC0152 48 h before infection. Scale bars represent 100 μm. (**D**) Acinar 266-6 cells were transfected with scrambled shRNA or STAT1-specific shRNA for 48 h and then treated with HJC0152 for 48 h before CVB3 infection. VP1 expression at 12 h p.i. was determined by Western blot assay. (**A**,**B**,**D**) GAPDH was probed as a protein loading control. The density of VP1 was first normalized to Histone H3 and then to the control. ns, no significance. *, *p* < 0.1; ***, *p* < 0.001. (**E**) CVB3 titer in culture supernatant was determined by plaque-forming assay. Quantification of plaque number per well is shown as mean ± SEM of averages from 3 independent experiments. ns, no significance. ***, *p* < 0.001.

**Figure 5 ijms-25-09007-f005:**
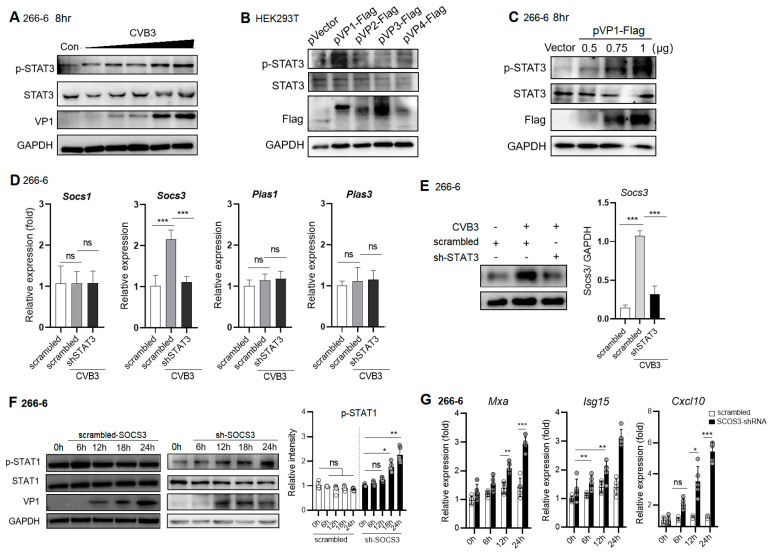
CVB3 VP1 protein activates STAT3 phosphorylation, which induces SCOS3 expression to inhibit STAT1. (**A**) Murine acinar 266-6 cells were infected with increased doses of CVB3 (MOI = 0.1, 1, 2, 5, 10). Cell lysates at 8 h p.i. were subjected to immunoblotting analysis of STAT3. (**B**) HEK293T cells were transfected with 1 μg pVP1-Flag, pVP2-Flag, pVP3-Flag or pVP4-Flag plasmid for 48 h, and then cell lysates were subjected to immunoblotting analysis to detect p-STAT3. (**C**) Immunoblot analysis of p-STAT3 in 266-6 cells transfected with 0.5, 0.75 and 1 μg pVP1-Flag for 48 h. (**D**,**E**) Acinar 266-6 cells were transfected with scrambled shRNA or STAT3-specific shRNA for 48 h before CVB3 infection (MOI = 1). Relative expressions of Socs1, Socs3, Pias1 and Pias3 (**D**) were detected by q-PCR. SOCS3 protein expression at 12 h p.i. was determined by Western blot assay (**E**). (**F**,**G**) Acinar 266-6 cells were transfected with scrambled shRNA or SOCS3-specific shRNA for 48 h before CVB3 infection (MOI = 1). Cell extracts were analyzed for levels of p-STAT1 and CVB3 VP1 using anti-pSTAT1 and anti-VP1 in Western blots (**F**). Quantification of relative pSTAT1 band intensities is shown. The relative mRNA expressions of Mxa, Isg15 and Cxcl10 were detected by Q-PCR (**G**). Data represent mean ± SEM (*n* = 5) of three independent experiments. ns, not significant. *, *p* < 0.1; **, *p* < 0.01; ***, *p* < 0.001.

**Figure 6 ijms-25-09007-f006:**
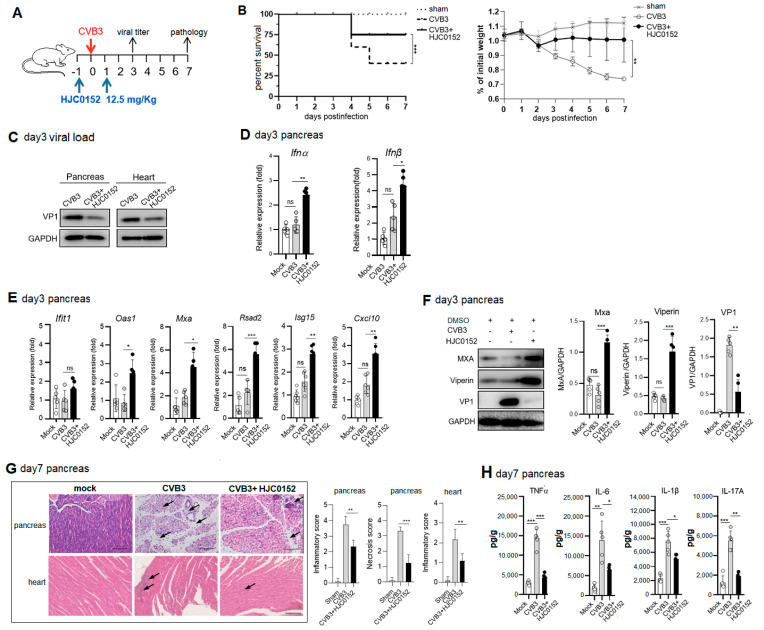
Therapeutic injection of STAT3 inhibitor reduces viral replication and AP and VMC pathology in mice. (**A**) Schematic map of HJC0152 treatment experiment. Mice were treated i.p. with 12.5 mg/Kg HJC0152 24 h before and 24 h after 103 pfu CVB3 infection. (**B**) Survival curve and weight loss curve in STAT3 inhibitor (HJC0152)-treated mice were followed by 7 dpi. (**C**) Immunoblotting analysis of day 3 pancreas or heart total lysates for VP1 expression. (**D**,**E**) Total RNAs from CVB3-infected day 3 pancreases were subjected to RT-QPCR using primers for IFN-α, IFN-β (**D**), ISGs (lfit1, Oas1, Mxa, Rsad2, Isg15 and Cxcl10) (**E**) and GAPDH. Relative mRNA was calculated by normalizing the values of the indicated genes to that of GAPDH. (**F**) Protein levels of MxA and Viperin from day 3 pancreases were quantitated by densitometric analysis and normalized to GAPDH, and they are presented as fold changes compared with the control. (**G**) Representative hematoxylin–eosin (H&E)-stained pancreas and heart sections from infected mice. Arrows indicate lymphocyte infiltration. Scale bar: 100 µm. (**H**) IL-1β, IL-6, TNF-α and IL-17A levels in day 7 pancreas homogenates were measured by ELISA. Data of (**C**–**H**) are represented as mean ± SEM (*n* = 5) of three independent experiments. ns, not significant. ***, *p* < 0.001; **, *p* < 0.01; *, *p* < 0.05.

## Data Availability

Data are contained within the article.

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
