# Peer review of "STAT3 Increases CVB3 Replication and Acute Pancreatitis and Myocarditis Pathology via Impeding Nuclear Translocation of STAT1 and Interferon-Stimulated Gene Expression"

_ijms, 2024, doi:10.3390/ijms25169007_

Round 1

Reviewer 1 Report

Comments and Suggestions for Authors

Certain statements in the Introduction need to have references/citations, such as L59, L79-83, L90-92, L96-97, L106-108, and sentences L227-229L328-329 require to clarify.

Could you please clarify why some virus names are in bold type or italics while others are not? This will help us understand the significance of the formatting and enhance our knowledge in this area.

The number of mice tested is not mentioned in Fig.1, and the pancreas and hearts are not indicated in Fig.1B. The results of hearts are missing Fig.2CD.

The format of reference is incorrect.

Comments on the Quality of English Language

Certain sentences are too long to understand or require to clarify. For instance, L12-15, L68-70, L79-83, L328-330.

Author Response

IJMS-3115351

Reviewer 1

  1. Certain statements in the Introduction need to have references/citations, such as L59, L79-83, L90-92, L96-97, L106-108, and sentences L227-229L328-329 require to clarify.

Respond: Very sorry for the references insufficiency. The relevant references have been supplemented in the revised manuscript. Long sentences have been simplified and clarified.

â‘  L59: References have been supplemented as :

[7] Schoggins, J.W. Recent advances in antiviral interferon-stimulated gene biology. F1000Res 2018, 7, 309.

[8] Haller O, Staeheli P, Schwemmle M, et al. Mx GTPases: dynamin-like antiviral machines of innate immunity. Trends. Microbiol. 2015, 23, 154-163.

[9] Shiratori, T.; Imaizumi, T.; Hirono, K.; Kawaguchi, S.; Matsumiya, T.; Seya, K.; Tasaka, S. ISG56 is involved in CXCL10 expression induced by TLR3 signaling in BEAS-2B bronchial epithelial cells. Exp. Lung. Res. 2020, 46, 195-202. 

[10] Dzimianski, J.V.; Scholte, F.E.M.; Bergeron, É.; Pegan, S.D. ISG15: It's Complicated. J. Mol. Biol 2019, 431, 4203-4216. 

[11] Panayiotou, C,; Lindqvist, R.; Kurhade, C.; Vonderstein, K.; Pasto, J.; Edlund, K.; Upadhyay, A.S.; Överby, A.K. Viperin Restricts Zika Virus and Tick-Borne Encephalitis Virus Replication by Targeting NS3 for Proteasomal Degradation. J Virol 2018, 92, e02054-17.

â‘¡L79-83:

“During the initial 7 days of CVB3 infection, p-STAT1 was found in heart of mice and mainly stained positive in only a few infiltrating immune cells within the infiltrated clusters [19]; IFNβ expression was below detection levels in the pancreas and increased in hearts of mice at 8 days p.i [16]. TLR3-PAR-1 cross-talk in mouse cardiac fibroblasts enhanced induction of IFN-β, p-STAT1 and CXCL10 expression upon CVB3 infection[20].”

[16] Schultheiss, H.P.; Rauch, U. Protease-activated receptor-2 regulates the innate immune response to viral infection in a coxsackievirus B3-induced myocarditis. J. Am. Coll. Cardiol. 2013, 62, 1737-1745.

[19] Ruppert, V.; Meyer, T.; Pankuweit, S.; Jonsdottir, T.; Maisch, B. Activation of STAT1 transcription factor precedes up-regulation of coxsackievirus-adenovirus receptor during viral myocarditis. Cardiovasc Pathol. 2008, 17, 81-92.

[20] Antoniak, S.; Owens, A.P. 3rd.; Baunacke, M.; Williams, J.C.; Lee, R.D.; Weithäuser, A.; Sheridan, P.A.; Malz, R.; Luyendyk, J.P.; Esserman, D.A., et al. PAR-1 contributes to the innate immune response during viral infection.. J Clin Invest. 2013, 123, 1310-22.

â‘¢L90-92, L96-97.

“STAT3, downstream of cytokine (IL-6, IL-10) or growth factor receptors, counteracts inflammation and promotes cell survival/proliferation and immune tolerance; while STAT1 inhibits proliferation and favors innate and adaptive immune responses[21] . STAT1 and STAT3 activation are reciprocally regulated and perturbation in their balanced expression or phosphorylation levels may re-direct cytokine/growth factor signals from inflammatory to anti-inflammatory[22]. In pathological contexts, IL-10-mediated STAT activation pattern in high IFNg/IFNa-stimulated macrophages switches from being predominantly STAT3 to be mainly STAT1, leading to increased STAT1-STAT3 heterodimers, strengthening the proinflammatory functions of IL-10 [23, 24].”

[21] Butturini, E.; Carcereri de Prati, A.; Mariotto, S. Redox Regulation of STAT1 and STAT3 Signaling. Int. J. Mol. Sci. 2020, 21, 7034. 

[22] Regis, G.; Pensa, S.; Boselli, D.; Novelli, F.; Poli, V. Ups and downs: the STAT1:STAT3 seesaw of Interferon and gp130 receptor signalling. Semin. Cell. Dev. Biol. 2008, 19, 351-359. 

[23] Sharif MN, Tassiulas I, Hu Y, Mecklenbräuker I, Tarakhovsky A, Ivashkiv LB. IFN-alpha priming results in a gain of proinflammatory function by IL-10: implications for systemic lupus erythematosus pathogenesis. J Immunol. 2004, 172, 6476-81.  

[24] Herrero, C.; Hu, X.; Li, W.P.; Samuels, S.; Sharif, M.N.; Kotenko, S.; Ivashkiv, L.B. Reprogramming of IL-10 activity and signaling by IFN-gamma. J Immunol. 2003, 171, 5034-5041. 

â‘£ L106-108:

“We speculate that over-activation of p-STAT3 in CVB3-infected pancreas and heart may function as a negative regulator of type I IFN response via inducing dysfunctional STAT1 activation, and weak ISGs production. ” That’s our speculation (scientific hypothesis) based on our data, no references could be supplemented.

⑤ sentence L227-229

The sentence has been clarified as : “To determine whether STAT3 inhibition promotes viral clearance, lentivirus-vectored shSTAT3 was transfected into 266-6 cells 48h before CVB3 infection, achieving 90% STAT3-inhibiting efficiency. Compared to scrambled shRNA, shSTAT3 significantly reduced VP1 expression (Fig. 4A). When 266-6 cells were treated with STAT3 inhibitor, HJC0152 or stattic, before CVB3 infection, both STAT3 inhibitors dose-dependently decreased p-STAT3 and VP1 expression (Fig. 4B). Immunofluorescence (IF) assay revealed that 12h after eGFP-CVB3(MOI=1) infection, HJC0152 markedly reduced eGFP+CVB3 progeny production compared to DMSO treatment (Fig. 4C)” 

â‘¥ Sentence L328-329

The sentence has been clarified as : “ STAT3 has a protective role at the early stage of several viral infections. STAT3 is required for optimal IFN response, splenic expression of IFN-α and ISGs to Herpes Simplex Virus-1 (HSV-1); Myeloid-specific STAT3 KO mice were more susceptible to HSV-1, as marked by higher viral loads and impaired NK and CD8+T cell activation [47] . Enterovirus 71 (EV71)-induced p-STAT3 exerts an anti-EV71 activity [48]. STAT3 exhibits antiviral activity in Influenza Virus (IAV) infection by activating ISGs (PKR, OAS2, ISG15, and MxB) expression [49]. However, more and more evidence show that STAT3 activation is associated increased viral replication. The epidermal growth factor receptor (EGFR) inhibitor, erlotinib, inhibites Hepatitis B virus (HBV) replication via downregulation of p-STAT3 in HBV-transfected HepG2.2.15 cells[50]. Influenza A virus infection activates STAT3 to enhance SREBP2 expression, cholesterol biosynthesis and virus replication [51]. Varicella-Zoster Virus (VZV) inhibits STAT1 activation, triggers p-STAT3 in vitro and in human skin xenografts in SCID mice in vivo; p-STAT3 inhibitors restrict VZV replication and skin malignant transformation[52]. Human Cytomegalovirus (HCMV) IE1 protein interacts with and promotes STAT3 nuclear accumulation which increases viral replication [53]. African swine fever virus (ASFV) CD2v activates the JAK2-STAT3 pathway in the early stage and inhibits apoptosis of infected cells, thereby promoting viral replication [54]. Furthermore, JAK2-STAT3 signaling contributes greatly to the cytokine storm seen in severe COVID 19 via inducing Th1/Th17 immune response and hyperactivating STAT3 but STAT1 dysfunction, thus suppressing anti-virus interferon response and coagulopathy [55]. ”

  1. Could you please clarify why some virus names are in bold type or italics while others are not? This will help us understand the significance of the formatting and enhance our knowledge in this area.

Respond: Very sorry for that format inconsistency. In our original submitted word version of the manuscript, all the virus names are in correct type and format. I don’t know either why the correct virus name format has been changed to that strange format after transformation into the pdf version (not by us, by the editor). We have corrected the format of all the virus names.

  1. The number of mice tested is not mentioned in Fig.1,

Respond: The number of mice (n=7) has been supplemented in the Fig 1 legend.

  1. the pancreas and hearts are not indicated in Fig.1B.

Respond: Thanks for your criticism. We have supplemented with the name of the tissues in revised Fig 1B.

  1. The results of hearts are missing Fig.2CD.

Thanks for your suggestion. Upon CVB3 infection of pancreas and hearts of mice, STAT3 was highly activated both in heart and pancreas. Although previous research has proven that STAT3 signaling is protective in CVB3-induced cardiac fibrosis; and cardiomyocyte-specific STAT3 depletion intensively affect physiological function of cardiomyocytes and ability to induce type I IFN response; however,in our experiment setting, in vivo (C57BL/6 mice) and in vitro (cardiomyocyte cell ling HL-1, acinar cell line 266-6), pre-treatment mice or cells with STAT3 inhibitor (HJC0152) before CVB3 infection, significantly reduced viral replication and titers withthin 3 days (or 24 h) of infection. And the initial viral lysis effect contribute greatly to tissue injury and inflammatory response. Especially in CVB3 infection of pancreas, the day3 pancreas display intensive necrotic cell death (30%~70% of the tissue) (Fig 1B) which is mainly due to direct viral lysis of acinar cells by CVB3. While on day3 p.i., the cardiomyocytes hardly exhibit cell lysis or necrotic injury. And the day3 viral titer in CVB3-infected pancreas is 2~3 log higher than that in the heart of mice. Therefore, we think the the effect and mechanism of hyper-activation of STAT3 in pancreas may differ from that of p-STAT3 in the heart. Therefore in this study, we mainly focus on the STAT3 activation and modulatory role of STAT3 on STAT1 translocation in the pancreas during acute CVB3 infection (0~7 days). And we provide data that CVB3 induced STAT3 hyper-activation in both pancreas and heart of mice on day3 p.i. and in acinar cell at 12h p.i. Effect of p-STAT3 on STAT1 expression and nuclear transportation and on SCOS3 expression are all studied in acinar cell line 266-6 cells.

In the current study, we provide the total p-STAT3, p-STAT1 expression in hearts and in HL-1 cells (Fig1E, 1F). And in revised Fig 2C, we supplemented with p-STAT3 and p-STAT1 expression in the nuclear & cytoplasmic fractions of day3 mouse heart, showing that CVB3 induced increased nuclear translocation of p-STAT3, but failed to induce nuclear export of p-STAT1 by Western Blot assay.

The STAT3 regulation STAT1 story in the heart is very important, however, could not be solved in the current study. We need much more fund and time to clarify the cardiac version of the story.  

  1. The format of reference is incorrect.

Respond: We are sorry for that. We have corrected the format of all the references.

  1. Certain sentences are too long to understand or require to clarify. For instance, L12-15, L68-70, L79-83, L328-330.

Respond: Thanks for your suggestion. We have modified all the long sentences.

L12-15,  the long sentence has been modified as “Up-regulation and protective role of STAT3 is reported in Coxsackievirus B3 (CVB3)-induced cardiac fibrosis; yet the exact role of STAT3 in modulating viral induced STAT1 activation and type I interferon (IFN)-stimulated gene (ISG) transcription in the pancreas remain unclarified. ”

L68-70, the long sentence has been modified as“Many viruses, including Respiratory syncytial virus (RSV) [12] and Severe Acute Respiratory Syndrome-Coronavirus-2 (SARS-CoV-2), have evolved various countermeasures against host IFN system and particularly, STAT1 expression and function [13].”.

L79-83, the long sentence has been modified as “STAT1 activation was found in heart of mice during the initial 7 days of CVB3 infection, mainly stained positive in a few infiltrating immune cells within the infiltrated clusters[17]. ”

L328-330, the sentences has been clarified and modified as “STAT3 has a protective role at the early stage of several viral infections. STAT3 is required for optimal IFN response, splenic expression of IFN-α and ISGs to Herpes Simplex Virus-1 (HSV-1); Myeloid-specific STAT3 KO mice were more susceptible to HSV-1, as marked by higher viral loads and impaired NK and CD8+T cell activation [47].Enterovirus 71 (EV71)-induced p-STAT3 exerts an anti-EV71 activity [48]. STAT3 exhibits antiviral activity in Influenza Virus (IAV) infection by activating ISGs (PKR, OAS2, ISG15, and MxB) expression [49]. However, more and more evidence show that STAT3 activation is associated increased viral replication. The epidermal growth factor receptor (EGFR) inhibitor, erlotinib, inhibites Hepatitis B virus (HBV) replication via downregulation of p-STAT3 in HBV-transfected HepG2.2.15 cells[50]. Influenza A virus infection activates STAT3 to enhance SREBP2 expression, cholesterol biosynthesis and virus replication [51]. Varicella-Zoster Virus (VZV) inhibits STAT1 activation, triggers p-STAT3 in vitro and in human skin xenografts in SCID mice in vivo; p-STAT3 inhibitors restrict VZV replication and skin malignant transformation[52]. Human Cytomegalovirus (HCMV) IE1 protein interacts with and promotes STAT3 nuclear accumulation which increases viral replication [53]. African swine fever virus (ASFV) CD2v activates the JAK2-STAT3 pathway in the early stage and inhibits apoptosis of infected cells, thereby promoting viral replication [54]. Furthermore, JAK2-STAT3 signaling contributes greatly to the cytokine storm seen in severe COVID 19 via inducing Th1/Th17 immune response and hyperactivating STAT3 but STAT1 dysfunction, thus suppressing anti-virus interferon response and coagulopathy [55].  ”

Reviewer 2 Report

Comments and Suggestions for Authors

In the manuscript entitled "STAT3 increases CVB3 replication and acute pancreatitis and myocarditis pathology via impeding nuclear translocation of STAT1 and interferon-stimulated gene expression", the authors demonstrate that infection with Coxsackievirus B3 (CVB3) impairs STAT3 regulated viral-induced STAT1 translocation, and subsequently IFN expression. The molecular mechanism how STAT3 influences IFN-I production (PMID: 16571725, PMID: 31293595, and many more) and the protective role of STAT3 in CVB3-infected cardiomyocytes (PMID: 33658937) has been described elsewhere, the novelty of the manuscript is low. Overall, the study might be of interest; however, there are some concerns that I have outlined below.

1) The authors should investigate the precise mechanism how CVB3 infection increases nuclear import of p-STAT3 but blunts nuclear translocation of p-163 STAT1, elucidating the precise mechanism would enhance the study's narrative.

2) It's recommended that the authors employ another technique, such as nuclear and cytoplasmic fractionation followed by direct measurement of p-STAT1 and p-STAT3 protein amounts under various conditions.

3) Identification of the viral protein that promotes activation of STAT3 will help to unravel new evasion strategies.

4) Please improve the image quality of IF staining.

5) Different IFN-alpha subtypes have different antiviral potential (depending on the virus type) (PMID: 31826721) – are all alpha subtypes affected ?

The presented manuscript is covering an interesting topic, and might be suitable for publication after addressing the following concerns. I hope that the authors can provide a revised manuscript addressing my concerns.

Comments on the Quality of English Language

moderate language editing is required

Author Response

Overall, the study might be of interest; however, there are some concerns that I have outlined below.

1)The authors should investigate the precise mechanism how CVB3 infection increases nuclear import of p-STAT3 but blunts nuclear translocation of p-163 STAT1, elucidating the precise mechanism would enhance the study's narrative.

Respond: Thank your very much for your comment.

â‘ We have screened the 4 CVB3 capsid proteins (VP1, VP2, VP3, VP4) for their ability to induce STAT3 phosphorylation in HEK293T cells after transfection of p-VP1/VP2/VP3/VP4-Flag plasmid before CVB3 infection, and identified VP1 as the capsid protein that induced p-STAT3 activation. And VP1 dose-dependently induced p-STAT3 expression in acinar cell 266-6 cells (as shown in revised new Fig 5A, B, C) .

Revised Fig 5A, B, C

â‘¡ As for the mechanism by which CVB3 increased nuclear import of p-STAT3 blunts nuclear translocation of p-STAT1, we adopted a series of experiments to screen out the negative regulatory gene induced by p-STAT3 that modulate STAT-1 activation. And we found SOCS3 mediated the inhibitory effect of STAT3 on p-STAT1. The new data (revised new Fig 5D-G) and the new Result 5 have been supplemented in the revised manuscript.

3.5 CVB3 VP1 induces STAT3 activation which induces expression of negative regulator, SOCS3 to block pSTAT1 /ISG expression

We confirmed that CVB3 dose-dependently promoted p-STAT3 expression in 266-6 cells (Figure 5A). To identify the viral capsid protein involved in triggering the activation of STAT3, we transfected HEK293T cells with plasmids encoding viral capsid protein VP1-VP4 48h before CVB3 infection and screened for their ability to induce p-STAT3 using Western Blot assay. Compared to vector-treated cell, of all these proteins, over-expressed VP1 markedly upregulated p-STAT3 expression (Figure. 5B). To confirm this effect, 0.5~1μg pVP1-flag was transfected into 266-6 cells 48h before CVB3 infection and it was found that VP1 protein dose-dependently increased p-STAT3 activation (Figure. 5C). Taken together, CVB3 capsid protein VP1 directly activates STAT3 activation in pancreas at an early phase of infection.

Distinct genes belonging to the Suppressor of Cytokine Signaling (SOCS) family are induced as immediate early genes downstream of STATs and inhibit STATs phosphorylation by different mechanisms in a negative-feedback loop [30]. p-STAT3 is associated with induction of the expression of SOCS3, a professional phosphorylation inhibitor of STATs [31]. Here, we wondered whether SOCS was involved in the inhibitory mechanism of p-STAT3 on p-STAT1. 266-6 cells were transfected with scrambled shRNA or shRNA-STAT3 48h before CVB3 infection and relative mRNA expression of STAT1 negative regulators, SOCS1, SOCS3 and protein inhibitor of activated STAT (PIAS) 1/3 (nuclear STAT binder) was evaluated. Compared to scrambled shRNA treatment, shRNA-STAT3 significantly increased mRNA expression of SOCS3 (Figure 5D), concomitantly with a robust induction of SOCS3 protein level (Figure 5E). To further assess the effect of SOCS3 in regulating virus-induced suppression of p-STAT1. Wild type (WT) and shRNA-SOCS3-transfected 266-6 cells were infected with CVB3. Cells lysates were assessed for p-STAT1. In contrast to no STAT1 activation in CVB3-infected WT cells, infection of SOCS3 knockdown cells resulted in strongly elevated p-STAT1 (Figure 5F). To answer the question whether enhanced p-STAT1 in SOCS-3 knockdown cells would also lead to enhanced expression of ISGs, total RNA was isolated at different time points p.i. from infected cells and monitored for induction of ISGs, Mxa, Isg15 and Cxcl10 (Figure 5G). mRNA levels of all three representative ISGs were elevated in SOCS3 knockdown versus WT cells at almost every time point during the course of infection. Meanwhile, knock down of SOCS3 resulted in decreased viral replication (Figure 5F). Taken together, the data indicate that in the absence of SOCS3, infection leads to a stronger activation of STAT1, resulting in enhanced expression of ISGs and reduced virus propagation.

 New Fig 5

Figure 5. CVB3 VP1 protein activates STAT3 phosphorylation which induced SCOS3 expression to inhibit STAT1.  (A) Murine acinar cell 266-6 were infected with increased doses of CVB3 (MOI=0.1, 1, 2, 5, 10). Cell lysates at 8h p.i. were subjected to immunoblotting analysis of STAT3. (B) HEK293T cells were transfected with 1 μg pVP1-flag, pVP2-Flag, pVP3-Flag or pVP4-Flag plasmid respectively for 48 hrs, then cell lysates were subjected to immunoblotting analysis to detect p-STAT3. (C) Immunoblot analysis of p-STAT3 in 266-6 cells transfected with 0.5, 0.75 and 1μg p-VP1-Flag for 48 hrs. (D-E) 266-6 cells were transfected with scrambled shRNA or STAT3-specific shRNA for 48 h before CVB3 infection (MOI=1). Relative expression of Socs1, Socs3, Pias1 and Pias3 (D) were detected by q-PCR. SOCS3 protein expression at 12 h p.i was determined by Western blot assay (E). (F-G) 266-6 cells were transfected with scrambled shRNA or SOCS3-specific shRNA for 48 h before CVB3 infection (MOI=1). Cell extracts were analyzed for levels of p-STAT1 and CVB3 VP1 using anti-pSTAT1 and anti-VP1 in Western blots (F). Quantification of relative pSTAT1 band intensities was shown. And the relative mRNA expression of Mxa, Isg15 and Cxcl10 is detected by Q-PCR (G). Data of represent as mean±SEM (n=5) of three independent experiments. ns, not significant. ***, p<0.001.

2) It's recommended that the authors employ another technique, such as nuclear and cytoplasmic fractionation followed by direct measurement of p-STAT1 and p-STAT3 protein amounts under various conditions.

Respond: Thanks for your suggestion. We had done the experiment you mentioned. Result 2.2 and Figure 2A, B, C are all data from immunoblotting of nuclear and cytoplasmic fractionation from infected pancreas, heart of mice or from infected acinar 266-6 cells. 

3) Identification of the viral protein that promotes activation of STAT3 will help to unravel new evasion strategies.

Respond: Thanks for you suggestion. According to your opinion, we transfected HEK293T cells with p-VP1-Flag, p-VP2-Flag, p-VP3-Flag and p-VP4-Flag plasmids respectively for 48h, then infected cells with CVB3 and screened the viral capsid protein that specifically promotes STAT3 activation. As shown in revised new Figure 5A-C, “We confirmed that CVB3 dose-dependently promoted p-STAT3 expression in 266-6 cells (Figure 5A). To identify the viral capsid protein involved in triggering the activation of STAT3, we transfected HEK293T cells with plasmids encoding viral capsid protein VP1-VP4 48h before CVB3 infection and screened for their ability to induce p-STAT3 using Western Blot assay. Compared to vector-treated cell, of all these proteins, over-expressed VP1 markedly upregulated p-STAT3 expression (Figure. 5B). To confirm this effect, 0.5~1μg pVP1-flag was transfected into 266-6 cells 48h before CVB3 infection and it was found that VP1 protein dose-dependently increased p-STAT3 activation (Figure. 5C). Taken together, CVB3 capsid protein VP1 directly activates STAT3 activation in pancreas at an early phase of infection.”

New Fig 5A, B, C

4) Please improve the image quality of IF staining.

Respond: We have improved the image quality of IF staining.

5) Different IFN-alpha subtypes have different antiviral potential (depending on the virus type) (PMID: 31826721) – are all alpha subtypes affected ?

Respond: Thank you very much for your valuable comment.

Very low levels of type I Interferons are induced in vitro and in hearts and pancreas of mice upon CVB3 infection. Previous studies have demonstrated that administration of murine IFN-β or polyriboinosinic: polyribocytidylic acid copolymer (pI:pC) at -24, 0, or 24 h p.i. significantly reduced CVB3 titers in mice and in vitro (Lutton CW, Gauntt CJ. Ameliorating effect of IFN-beta and anti-IFN-beta on coxsackievirus B3-induced myocarditis in mice. J Interferon Res. 1985 Winter;5(1):137-46. ). Recent study indicates that CVB3 induced relative (to pancreas) abundant levels of IFNa and IFNβ in liver and hepatocytes; IFNAR KO or hepatocyte-specific IFNAR KO mice experienced severe liver necrosis and increased viral titer after CVB3 infection (Koestner W, Spanier J, Klause T, et al. Interferon-beta expression and type I interferon receptor signaling of hepatocytes prevent hepatic necrosis and virus dissemination in Coxsackievirus B3-infected mice. PLoS Pathog. 2018 Aug 3;14(8):e1007235. ). Mechanically, IFN-β regulates glucose metabolism (decrease p-AMPK, increase in intracellular ATP) and important for antiviral activity against CVB3 (Burke JD, Platanias LC, Fish EN. Beta interferon regulation of glucose metabolism is PI3K/Akt dependent and important for antiviral activity against coxsackievirus B3. J Virol. 2014 Mar;88(6):3485-95. ).

Treatment with mIFN-b [2.5 to 10 million international units (MIU)/kg] dose-dependently eliminated cardiac viral load and improved the general health status in CVB3-inoculated mice. Notably, Treatment with 10 MIU/kg mIFN-a(2) resulted in a similar level of efficacy as that induced by 5 MIU/kg mIFN-β, with the exception that mIFN-a(2) did not reduce cardiac CVB3 mRNA. However, mIFN-a(2) significantly attenuated CVB3-induced epicarditis (Wang YX, da Cunha V, Vincelette J, et al. Antiviral and myocyte protective effects of murine interferon-beta and -{alpha}2 in coxsackievirus B3-induced myocarditis and epicarditis in Balb/c mice. Am J Physiol Heart Circ Physiol. 2007 Jul;293(1):H69-76. ).

Although IFN-I is known to decrease viral replication in viral infections, yet, during several persistent infections, chronic IFN-I production leads to pathogenic outcomes and disease progression. IFN-I signaling is essential for the expression of the negative immune regulators IL-10 and PD-L1 following persistent LCMV infection [Teijaro JR, Ng C, Lee AM, et al. Persistent LCMV infection is controlled by blockade of type I interferon signaling. Science. 2013 Apr 12;340(6129):207-11]. IFN-α subtypes have been implicated in many autoimmune conditions. Autoantibodies (aAbs) against IFN-I which are often present in autoimmune disorders are a serious risk factor for severe disease during viral infection. pDCs producing IFN-α and IL-33 play a pivotal role in driving the chronic fibro-inflammatory responses underlying human IgG4-related autoimmune pancreatitis (AIP) and murine AIP (Watanabe T, Yamashita K, Arai Y, et al. Chronic Fibro-Inflammatory Responses in Autoimmune Pancreatitis Depend on IFN-α and IL-33 Produced by Plasmacytoid Dendritic Cells. J Immunol. 2017 May 15;198(10):3886-3896.).

In patients with type 1 diabetes, and in mice infected with CVB3 or CVB4, non-neutralizing antibodies can increase the infection of monocytes and stimulate the production of IFN-α by these cells in vitro; resulting in an extended inflammatory reaction and myocarditis via antibody-dependent enhancement (ADE) of infection (Hober D, Sane F, Jaïdane H, et al. Immunology in the clinic review series; focus on type 1 diabetes and viruses: role of antibodies enhancing the infection with Coxsackievirus-B in the pathogenesis of type 1 diabetes. Clin Exp Immunol. 2012 Apr;168(1):47-51.).

Taken together, type I IFNs, particularly, IFNβ, has proved its important and critical treating effect against CVB3 in vitro and in vivo. While effects of IFN-a subtypes in CVB3 infection are moderate and might be associated with a risk to autoimmune disease.

Our study focuses on the modulatory effect and mechanism of activated p-STAT3 on poorly induced p-STAT1 signaling, IFNb/a and ISGs transcription. To study the induction and effects of various IFN-a subtypes on CVB3 replication in mice and in cells, we need extra financial support and much more time to solve the interesting problem.

6) The presented manuscript is covering an interesting topic, and might be suitable for publication after addressing the following concerns. I hope that the authors can provide a revised manuscript addressing my concerns.

moderate language editing is required.

Respond: Thanks for your encouraging. The language and grammar have been modified. 

.

Round 2

Reviewer 1 Report

Comments and Suggestions for Authors

The manuscript has been improved, and the questions have been answered.

No further comments.

Reviewer 2 Report

Comments and Suggestions for Authors

The authors have addressed all my concerns. No further issues detected.